# The effects of locomotor activity on gastrointestinal symptoms of irritable bowel syndrome among younger people: An observational study

Toyohiro Hamaguchi[1,2]*, Jun Tayama[2,3], Makoto Suzuki[4], Naoki Nakaya[1,2], Hirokazu Takizawa[1], Kohei Koizumi[1], Yoshifumi Amano[1], Motoyori Kanazawa[3], Shin Fukudo[3]

1 Department of Rehabilitation, Graduate School of Health Science, Saitama Prefectural University, Saitama, Japan, 2 Department of Behavioral Medicine, Graduate School of Medicine, Tohoku University, Sendai, Japan, 3 Faculty of Human Sciences, Waseda University, Saitama, Japan, 4 Faculty of Health Sciences, Tokyo Kasei University, Saitama, Japan

* hamaguchi-toyohiro@spu.ac.jp

**Data Availability Statement:** All relevant data are within the paper and its Supporting Information files.

## Abstract

Irritable bowel syndrome (IBS) is a common bowel disorder that manifests as unexplained abdominal pain or discomfort and bowel habit changes in the form of diarrhea, constipation, or alternating patterns of the two. Some evidences demonstrate that increased physical activity improves IBS symptoms. Hence, daily exercise is recommended in these patients. In this study, we aimed to investigate the relationship between physical activity and gastrointestinal symptoms in 101 university students (female = 78) with IBS. Participants were examined by Gastrointestinal Symptoms Rating Scale (GSRS), and gait steps were measured for 1 week using a pedometer. The association between the GSRS score and pedometer counts was determined by ordinal logistic modeling analysis. The ordinal logistic regression model for GSRS and locomotor activity showed a significant stepwise fit ($z =$ -3.05, $p = 0.002$). The logistic curve separated GSRS score of 5 points (moderately severe discomfort) from 2 points (minor discomfort) by locomotor activity. The probability for daily locomotor activity to discriminate between 5 and 4 points of GSRS (i.e., likely to have reverse symptoms) decreased in accordance with increment of steps per day: 78% probability for 4000 steps, 70% probability for 6000 steps, 59% probability for 8000 steps, and 48% probability for 10000 steps. This study demonstrated that the severity of GSRS is associated with the amount of walking in younger people with IBS. These results may be used as a measure to determine the daily step count to reduce the severity of gastrointestinal symptoms in individuals with IBS.

## Introduction

Irritable bowel syndrome (IBS) is a common bowel disorder that manifests as unexplained abdominal pain or discomfort and bowel habit changes in the form of diarrhea, constipation,

**Funding:** The Japan Society for the Promotion of Science funded this research through grants (project JP10K11368 and JP18KK0275). Staff at Saitama Prefectural University provided support for this study through subject recruitment, material procurement, and funding management.

**Competing interests:** The authors have declared that no competing interests exist.

or alternating patterns between the two [1, 2]. IBS is associated with reduced quality of life [3, 4], which may affect every day living activities. IBS is reportedly common in Japanese adults and is more prevalent in younger age females with low body mass index (BMI) [5]. We followed the treatment guidelines developed by the Japanese committee of Gastroenterology [6]. As some patients prefer non-medical management through diet and physical activity [7, 8], lifestyle modification is considered the initial management for IBS. A previous study has shown that increased physical activity improves gastrointestinal (GI) symptoms in patients with IBS [9]. In the National Health Promotion Movement in the 21st Century in Japan (Health Japan 21), the recommended amount of daily activity is equivalent to 9000 steps for men and 8500 steps for women aged 20–64 years [10]. Thus, it is desirable to achieve these targets in IBS patients.

Previous studies have reported that after a 12-week intervention, GI symptoms improved in physically active patients with IBS compared with physically inactive patients [7, 11]. Brief yoga poses and breathing intervention were feasible and safe adjunctive treatments in young patients with IBS, leading to reduced pain and GI symptoms [12]. Moreover, the IBS group had significantly improved symptoms of constipation compared to the usual care group at a 12-week follow-up after exercise [9]. Hence, physical exercise may be effective in attenuating IBS symptoms temporarily.

Adolescents reportedly present with a variety of IBS symptoms [13, 14] and physical activities [15, 16]. If the relationship between the severity of IBS symptoms and the amount of physical activities can be clarified, then the minimum amount of daily exercise to reduce IBS symptoms can be determined. However, to the best of our knowledge, no study has reported on the extent of the effects of daily activity on improving the GI symptoms of IBS. The research question of how activity level influences symptom severity is important to the management of IBS symptoms. Thus, we aimed to investigate the relationship between physical activity and GI symptoms among younger people with IBS and to estimate the extent of GI symptoms attenuated by achieving the recommended amount of daily activity as per Health Japan 21 [10].

## Methods

### Study design and ethical considerations

This was an observational study. This study was approved by the Ethics Committee in Saitama Prefectural University (No. 27157) and was conducted in accordance with the Declaration of Helsinki. Participants were informed that the purpose of this study was to investigate the relationship between GI symptoms and physical activity among participants with IBS during the informed consent procedure. Furthermore, they were instructed to measure their physical activity for 1 week using a Pedometer (LifeCorder GS, Suzuken, Tokyo), and to wear the pedometer for 1 week (except when taking a bath), and also to perform their daily life activities.

### Participants

The inclusion criteria for study participants were as follows: (1) university students >20 years old and (2) students diagnosed with IBS symptoms according to Rome III criteria: recurrent abdominal pain or discomfort associated with two or more of the following: 1) improvement with defecation; and/or, 2) onset associated with a change in frequency of stool; and/or 3) onset associated with a change in form (appearance) of stool for at least 3 months in the last 6 months [17]. The exclusion criteria were (1) students taking medication for IBS treatment 12 weeks prior to the start of the study and (2) students in whom locomotor counts for 1 week could not be measured. The number of participants required for the analysis of this study was

67 as calculated by G* power [18], logistic regression a priori with an effect size of 0.8, an alpha error of 0.05, and a power of 0.8.

## Data collection

From 2015 to 2018, we distributed 1240 copies of survey cooperation requests to university students annually between October to January. In this study, university students were recruited to investigate the relationship between IBS symptoms and physical activity, from autumn to winter. A request for recruiting collaborators in the survey was created, posted on the university bulletin board, and distributed to university students after class. A school medical doctor interviewed the students who read the distributed survey request form and confirmed the presence or absence of IBS symptoms according to the Rome III criteria.

The Rome III criteria are used to diagnose IBS symptoms, which include recurrent abdominal pain or discomfort, 3 days per month in the last 3 months (12 weeks), and are associated with two or more of the following three criteria: 1) improvement with defecation, 2) the onset is associated with a change in stool frequency, and 3) the onset is associated with a change in the stool form (appearance). To fulfil the criteria, symptom onset should occur 6 months prior to the diagnosis.

Informed consent forms were given to students who had IBS symptoms, and consent to measure the number of steps in 1 week and to investigate GI symptoms using the Gastrointestinal Symptoms Rating Scale (GSRS) [19, 20] was obtained. The GSRS is a disease-specific instrument of 15 items combined into 5 symptom clusters depicting reflux, abdominal pain, indigestion, diarrhea, and constipation. The GSRS has a seven-point graded Likert-type scale where "1" represents the absence of troublesome symptoms and "7" represents very troublesome symptoms.

Students responding to the survey carried pedometers (LifeCorder GS, Suzuken, Tokyo) about for 1 week, after which each individual's weekly walking activity and digestive symptom scores were analyzed. Walking activity data that were recorded in LifeCorder GS were uploaded into a personal computer using an application Lifelyzer05 (Kenz, Tokyo). Participant's GI symptoms were examined using the GSRS after pedometer counts.

## Statistical analysis

Participants' age, sex, physical activity, and GI symptoms were compared according to sex using χ2 test and Student's t-test. The GSRS scores were derived from the total score and divided by 15 (i.e. the 15 item subscales). The average pedometer counts (steps/day) were calculated using all days of data collection. The association between the GSRS score and pedometer counts was determined by the ordinal logistic modeling analysis [21]. The relationship between GSRS score and pedometer counts (prediction probability $g(x)$) was estimated using ordinal logistic regression modeling (Eq 1) with the dependent variable as GSRS score ($f(x)$, continuous variates 1 to 7) and the independent variable as pedometer counts for $x$ (Eq 2). The principle of ordinal logistic regression modeling is to fit the probability ($P$) of multiple dichotomous responses (Eq 1):

$$g(x) = \frac{1}{1 + e^{-f(x)}} \tag{1}$$

$$f(x) = \beta_0 + \beta_1 x + e \tag{2}$$

where $x$ is the explanatory variable, $\beta_i$ is the partial regression coefficient, and $e$ is the residual between actual and predicted data. Therefore, for multilevel ordinal responses, the cumulative

probability is calculated at each level to generate a simple regression. In this study, the probability of the cut-off point for each level of severity of GI symptoms based on the GSRS score (1|2, 2|3, 3|4, 4|5) was evaluated in association with the pedometer count. A sub-analysis was performed to investigate any gender difference. The level of statistical significance was set at 5%. All statistical analyses were performed using the R 3.5.2 software (R Foundation for Statistical Computing, Vienna, Austria).

## Results

Of 663 participants who consented to enter the study, 103 university students (80 female) were diagnosed by school medical doctors to have IBS symptoms based on the Rome III criteria [17, 22], none of whom were taking medications for IBS. Of the 103 participants, two were excluded before the analysis because the step counts were not measured every day. Finally, data from 101 participants (female = 78) were analyzed (Fig 1). IBS subtypes, based on the frequency of symptoms in the participants, were constipation (n = 42), mixed type (n = 29), diarrhea (n = 25), and not classified (n = 5). All 101 participants completed the one-week step count and GSRS survey. The number of female students with IBS was higher than that of male students ($x^2$ = 3.36, $p < 0.01$, V = 0.04). The BMI was 23 ± 3 for female participants and 21 ± 2 for male participants. No sex differences in age, one-week step count, and GSRS score were found (Table 1).

Scatterplots of GSRS score and locomotor activity of the participants are presented in Fig 2a. The ordinal logistic regression model for GSRS and locomotor activity showed a significant stepwise fit (z = -3.05, $p$ = 0.002; Fig 2b). The GSRS ranges from severe, to moderate, to minor discomfort. The threshold estimate assigned to severe is GSRS score of 5, to moderate GSRS score of 3 and to minor discomfort GSRS score of 2. Locomotor activity was a significant predictor in separating these thresholds with the estimate assigned to this logistic curve.

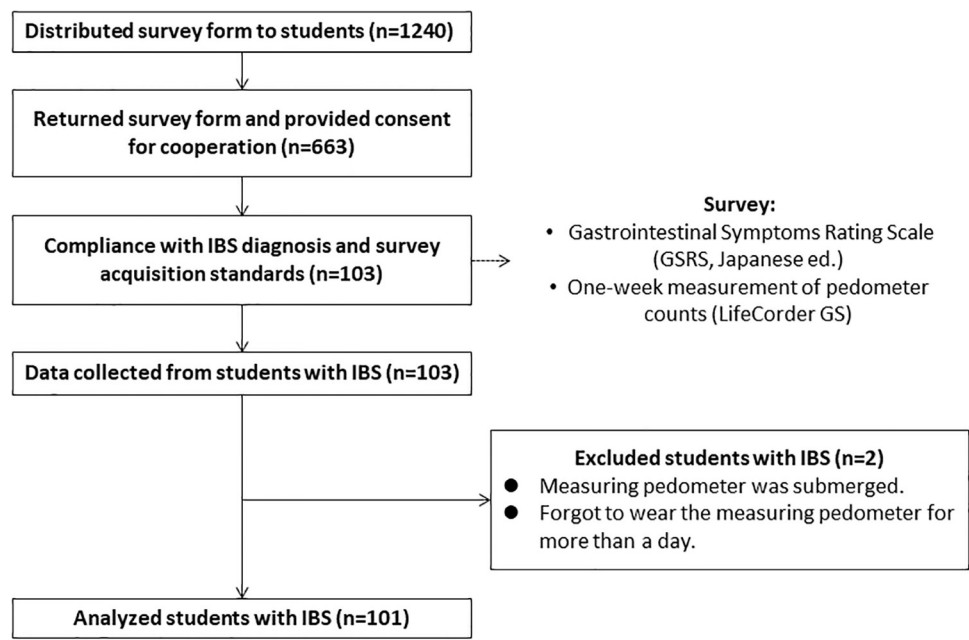

**Fig 1. Selection process of the study population and study design.** The survey was distributed to a total of 1240 university students during the study period. Data from 101 students who met the inclusion criteria were collected and statistically analyzed.

**Table 1. Participant characteristics.**

|  | All | Female | Male | Statistics | | |
|---|---|---|---|---|---|---|
| Participants (n) | 101 | 78 | 23 | $x^2 = 3.36$ | $p < 0.01$ | V = 0.12 |
| Age (years) | 20 ± 2 | 20 ± 2 | 20 ± 2 | t = -0.59 | $p = 0.56$ | d = 0.09 |
| Locomotor | 8126 ± 2570 | 7627 ± 2426 | 8272 ± 2608 | t = 1.37 | $p = 0.31$ | d = 0.26 |
| GSRS score | 2.6 ± .9 | 2.6 ± .9 | 2.7 ± .8 | t = -0.77 | $p = 0.44$ | d = 0.11 |

The locomotor activity is the number of daily step counts measured using a LifeCorder GS pedometer that participants carried for 1 week. The Gastrointestinal Symptoms Rating Scale (GSRS) score is adjusted by dividing the total score by the number of questions.

Probability for daily locomotor activity to discriminate between GSRS scores 5 and 4 (i.e., likely to have severe symptoms) was decreased in accordance with increment of steps per day: 78% probability for 4000 steps, 70% probability for 6000 steps, 59% probability for 8000 steps, and 48% probability for 10000 steps (Fig 2b and S1 Table).

Sub-analysis was performed separately to determine sex differences. We did not observe any significant difference in the results of the ordinal logistic regression analysis for the male participants (z = -1.81, $p = 0.07$), while that of female participants showed a significant difference (z = -2.44, $p = 0.01$), which was similar to the overall results of the ordinal logistic regression analysis (Fig 3). Probability for daily locomotor activity to discriminate between GSRS scores 5 and 4 was decreased in accordance with increment of steps per day: 79% probability for 4000 steps, 71% probability for 6000 steps, 62% probability for 8000 steps, and 52% probability for 10000 steps in female participants (Fig 3b). Especially, probability for daily locomotor activity to discriminate between GSRS scores 5 and 4 was 60% probability for 8500 steps per day in reference to necessary daily steps in healthy females recommended by the Health Japan 21 (S2 Table). [10]

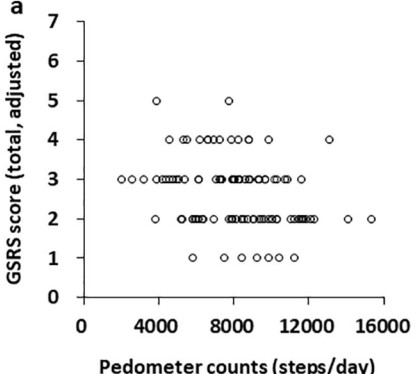
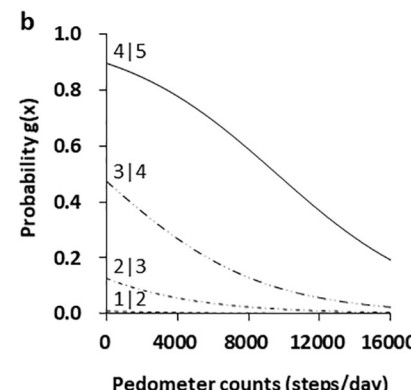

**Fig 2. Logistic probability plots of the relationship between GSRS score and pedometer counts.** (A) Scatterplots of Gastrointestinal Symptoms Rating Scale (GSRS) scores and daily pedometer counts in university students with IBS (n = 101). GSRS scores range from 7 indicating "very severe discomfort" to 1 "no discomfort at all." Plots were realigned by GSRS scores and the one-week pedometer counts in participants with irritable bowel syndrome (IBS). (B) Logistic curves separated by GSRS scores of 5 (moderately severe discomfort) and 2 (broken line: minor discomfort), GSRS scores of 5 and 4 (solid line), and GSRS scores of 4 and 3 (chain line) were in a stepwise fit. Ordinal logistic regression model, z = -3.05, stepwise fit $p = 0.002$.

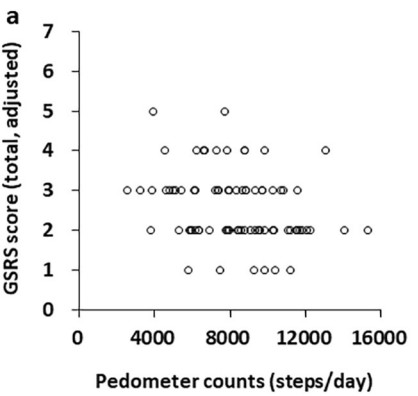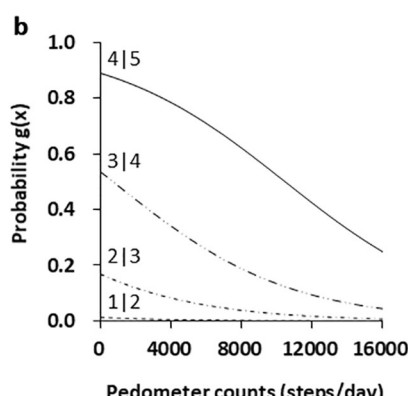

**Fig 3. Logistic probability plots of the relationship between GSRS score and pedometer counts in female participants.** (A) Scatterplots of Gastrointestinal Symptoms Rating Scale (GSRS) score and daily pedometer counts in female participants with irritable bowel syndrome (IBS) (n = 78). (B) Logistic curves separated by GSRS scores of 5 (moderately severe discomfort) and 2 (broken line: minor discomfort), GSRS scores of 5 and 4 (solid line), and GSRS scores of 4 and 3 (chain line) were in a stepwise fit. Ordinal logistic regression model, z = -2.44, stepwise fit $p$ = 0.01.

## Discussion

This study investigated the relationship between the number of daily step counts and GI symptoms by applying an ordinal logistic model to data collected from younger people with IBS. Our results indicated that locomotor activity and GI symptoms were correlated, and threshold levels of locomotor activity that could predict GI symptoms in IBS exist. We have also observed independent probabilities for IBS symptoms in relation to locomotor activity. Our findings suggest gender difference in the symptoms and its effects, which was predictable considering the female predominance of our cohort. On the ordinal logistic regression analysis data from the female participants were comparable to the overall results of the analysis, while the data from the male participants were not. Therefore, the results of this study can also serve as reference values for young female patients with IBS.

To the best of our knowledge, using a model formula to predict GI symptoms of IBS based on daily step counts is a novel method in this field. Therefore, we estimated the degree of attenuation of GI symptoms of IBS in our participants, especially in young females, by applying this formula to the recommended daily momentum in Healthy Japan 21 [10]. For example, probability for daily locomotor activity to discriminate between GSRS scores 5 and 4 was 60% probability for 8500 steps per day in reference to the recommended daily step counts for a healthy female according to the Health Japan 21. This result indicates that the equivalent number of steps recommended by Healthy Japan 21 may also be an effective target for patients with IBS.

Physical activity using a pedometer-based guideline could increase public health outcomes, [23, 24] with 3000 steps in 30 min (100 steps/min) considered as moderate-intensity activity [25]. Increased physical activity may improve IBS symptoms through different mechanisms [6, 8]. Mild physical activity enhances intestinal gas clearance and reduces symptoms in patients with bloating [26, 27]. To increase colon transit time in adults with chronic constipation [28], 30 min of daily walking is recommended to improve the defecation pattern [29]. A recent study demonstrated that inflammatory biomarkers [30] were attenuated after 24 weeks of moderate-intensity aerobic exercise [31]. Thus, routine physical activity may be a useful primary treatment modality in IBS.

This study has some limitations. (1) The participants in this study had mild IBS symptoms but were not taking medication. Since IBS improves with medications [32, 33] it is necessary

to investigate whether there is a relationship between locomotor activity and digestive symptoms among patients taking medications. (2) Dietary therapy is effective in IBS [34, 35]. However, we did not investigate patients' meal contents during the study period. It is unclear whether dietary contents affect the relationship between exercise and GI symptoms in IBS. Future studies are required to investigate such correlations. (3) This study did not use an index to estimate IBS symptoms other than GSRS. It is necessary to verify the relationship and effects using IBS-QOL [36] and other indicators. (4) The present analysis targeted younger people, and no stratification analysis was performed based on age. The recommended amount of locomotor activity differs between the young and elderly individuals in Health Japan 21 [10]. Thus, it is necessary to investigate the relationship between GI symptoms and daily effects of locomotor activity and to build a prediction model for elderly people. (5) There was no significant association between GI symptoms and locomotor activity in males with IBS in this study. There were gender differences in the symptoms of IBS [37] characterized by constipation and diarrhea. The Prevalence of IBS subtypes were similar to those in a previous study [38]; no IBS subtypes were included in our analysis. In addition, symptoms vary with age [39]; therefore, future studies with larger cohorts should be stratified by age and IBS subtype to further investigate the relationship between physical activity and digestive symptoms. (6) The purpose of the study was explained to the participants during the informed consent process. Thus, participants may have increased their physical activity during the study period since they were informed that IBS symptoms are affected by the amount of physical activity. It is unclear whether the information bias was influenced by the participants' knowledge of the study aims. Therefore, the effect of the bias should be verified by a study examining the effects of information on the relationship between GI symptoms and physical activity among people with IBS. (7) This study used Rome III criteria, when the study was planned in 2015. The IBS diagnostic criteria were updated to ROME IV [40, 41], and the Japanese version of the questionnaire has not yet been published. In Rome IV, the evaluation of the symptoms has changed from that of Rome III, including the emphasis on the subjective experience of abdominal pain in IBS patients, and the demarcation between the constipation subtype of IBS and functional constipation. In subsequent studies, IBS diagnostic criteria should be based on ROME IV.

Based on our findings, increasing the daily step count to 9500 steps from 4000 steps will result in 50% reduction in the severity of symptoms. Previous studies have shown that exercise improves IBS symptoms [7, 11]. Current data has suggested the "degree" (amount) of physical activity required to attenuate IBS symptoms. The results of this study can provide the clinicians with information on how many steps to add to the current physical activity level among IBS patients that can reduce GSRS by 1 point. However, the effect of exercise on symptom improvement in IBS patients with mild to moderate discomfort is considered small. The effects of locomotor activity in decreasing GSRS scores should be determined in an intervention study in the future. It is recommended that patients with IBS exercise on a daily basis; however, there is no consensus on the type of exercise to be performed. In conclusion, the results of this study demonstrated that the amount of locomotor activity was related to GI symptoms in younger people with IBS by applying an ordinal logistic model. Furthermore, the results suggest that the amount of daily locomotor activity may attenuate IBS symptoms among younger people, especially female IBS patients. These results may be used as a measure to determine the daily step counts for reducing the severity of GI symptoms in individuals with IBS.

## Supporting information

**S1 Table. Target values for daily step counts in younger people with IBS.** Estimated probability rate for Gastrointestinal Symptoms Rating Scale (GSRS) score by ordinal logistic

modeling in all participants in this study (n = 101). The Health Japan 21 recommended a daily activity level of 8500 steps/day for females and 9000 steps/day for males. IBS, irritable bowel syndrome.
(DOCX)

**S2 Table. Target values for daily step counts in younger females with IBS.** Estimated probability rate for Gastrointestinal Symptoms Rating Scale (GSRS) score by ordinal logistic modeling. The Health Japan 21 recommended a daily activity level of 8500 steps/day for females. If a female patient with IBS walked only 4000 steps/day, she will attain GSRS score 5 with probability of 78.5%, while 8500 steps/day will reduce the probability to 59.7%. IBS, irritable bowel syndrome.
(DOCX)

**S1 Dataset.**
(XLSX)

## Acknowledgments

The authors would like to thank all the staff of the department of Regional Industry-Academia Collaboration, Saitama Prefectural University (Koichi Suda, Akiko Yanagisawa, Keiko Hatano, Shigemi Wakisaka, Mina Takeuchi, Miho Kitada and, Masami Shirota) for their contributions and the operational approval to conduct this study.

## Author Contributions

**Conceptualization:** Toyohiro Hamaguchi, Jun Tayama, Shin Fukudo.

**Data curation:** Toyohiro Hamaguchi, Hirokazu Takizawa, Kohei Koizumi, Yoshifumi Amano.

**Formal analysis:** Toyohiro Hamaguchi, Makoto Suzuki, Hirokazu Takizawa, Kohei Koizumi, Yoshifumi Amano.

**Funding acquisition:** Toyohiro Hamaguchi, Jun Tayama.

**Methodology:** Toyohiro Hamaguchi, Jun Tayama, Kohei Koizumi, Motoyori Kanazawa, Shin Fukudo.

**Project administration:** Toyohiro Hamaguchi.

**Resources:** Hirokazu Takizawa, Yoshifumi Amano.

**Software:** Makoto Suzuki.

**Supervision:** Naoki Nakaya, Motoyori Kanazawa, Shin Fukudo.

**Validation:** Makoto Suzuki, Motoyori Kanazawa, Shin Fukudo.

**Visualization:** Toyohiro Hamaguchi, Makoto Suzuki.

**Writing – original draft:** Toyohiro Hamaguchi, Makoto Suzuki.

**Writing – review & editing:** Naoki Nakaya, Motoyori Kanazawa, Shin Fukudo.

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
