## [Decision Letter · Decision Letter 0]

18 Mar 2020

PONE-D-19-33098

The effects of locomotor activity on gastrointestinal symptoms of irritable bowel syndrome: an observational study

PLOS ONE

Dear Prof. Hamaguchi,

Thank you for submitting your manuscript to PLOS ONE. After careful consideration, we feel that it has merit but does not fully meet PLOS ONE’s publication criteria as it currently stands. Therefore, we invite you to submit a revised version of the manuscript that addresses the points raised during the review process.

Editor: The reviewer 1 has good points, please address them carefully. Some specific comments:

Please explain how each of the variables used in the analysis were measured.Statistical analysis should also cover the sub-analysis which is presented 

We would appreciate receiving your revised manuscript by May 02 2020 11:59PM. To enhance the reproducibility of your results, we recommend that if applicable you deposit your laboratory protocols in protocols.io, where a protocol can be assigned its own identifier (DOI) such that it can be cited independently in the future. For instructions see: http://journals.plos.org/plosone/s/submission-guidelines#loc-laboratory-protocols

We look forward to receiving your revised manuscript.

Kind regards,

Subas Neupane

Academic Editor

PLOS ONE

Journal Requirements:

2. Please provide additional details regarding participant consent. In the ethics statement in the Methods and online submission information, please ensure that you have specified what type of consent you obtained (for instance, written or verbal). If your study included minors, state whether you obtained consent from parents or guardians. If the need for consent was waived by the ethics committee, please include this information.

4.Thank you for stating the following financial disclosure:

"The funders had no role in study design, data collection and analysis, decision to

publish, or preparation of the manuscript."

a) Please provide an amended Funding Statement that declares *all* the funding or sources of support received during this specific study (whether external or internal to your organization) as detailed online in our guide for authors at http://journals.plos.org/plosone/s/submit-now. 

b) Please state what role the funders took in the study.  If any authors received a salary from any of your funders, please state which authors and which funder. If the funders had no role, please state: "The funders had no role in study design, data collection and analysis, decision to publish, or preparation of the manuscript."

Reviewers' comments:

Reviewer's Responses to Questions

**Comments to the Author**

1. Is the manuscript technically sound, and do the data support the conclusions?

Reviewer #1: No

Reviewer #2: Yes

2. Has the statistical analysis been performed appropriately and rigorously? 

Reviewer #1: N/A

Reviewer #2: Yes

3. Have the authors made all data underlying the findings in their manuscript fully available?

Reviewer #1: No

Reviewer #2: Yes

4. Is the manuscript presented in an intelligible fashion and written in standard English?

Reviewer #1: Yes

Reviewer #2: Yes

5. Review Comments to the Author

Reviewer #1: 1. The question of how activity level influences symptom severity is important to the management of IBS symptoms.

2. There were 1240 surveys distributed and 663 (53%) returned usable data. 103 of these had an IBS diagnosis and were included in the study. It is not clear whether the participation rate was influenced by the subject's knowledge of the study aims.

3.. Sex was predominantly female, with 78 females to 23 males. Sex was not a significant predictor. Should it be included in the analysis?

4. No instructions to subjects are included in the manuscript. These may influence the outcome of the data.

5. Among the predictor variables (age, sex, symptom severity, and activity), only activity level was significantly related to symptom severity. Age was restricted because the population were students. Although none of the predictor variables was related to symptom severity, an "ordinal logistic regression" method was used to control for overlapping classification of the predictors. The logic is not clear to me and should be explained. The supporting tables suggest that the data were analyzed by changes in activity level needed to move a subject from one severity classification to another on a 5-point scale. If this method appropriately controls for the classification overlap, the analysis shows a strong effect of age and sex on the severity of IBS symptoms.

6. There are some limitations to the study which should be explained: Age was limited by studying students. Sex is predominantly female; should the males be included in the analysis? Students receiving medications for IBS were excluded; does this bias the outcome?

7. Explain how this study extends previous studies of the effects of physical activity on IBS symptom severity. How should clinicians use the findings to manage patients?

8. Please explain the tables and figures. Are changes in activity level related to the baseline level of activity? Are the recommended changes achievable?

Reviewer #2: The aspect that the study addresses is important. Although the effect exercise on IBS is well documented, this study provides a quantitative measure of effect of walking on IBS symptoms. It will be helpful for management of this disease. The manuscript is well written. I have one main question.

What was the reason the authors used Rome III criteria for confirming IBS instead of Rome IV criteria? Please see the following three references given as example how using criteria Rome IV versus Rome III impacts studies.

(i) Am J Gastroenterol (2018) 113:1017–1025. https://doi.org/10.1038/s41395-018-0074-z

(ii) Lin and Chang; Clinical Gastroenterology and Hepatology Vol. 18, No. 2

(iii) https://doi.org/10.1111/nmo.13189

The authors need to address this in the manuscript as it may impact the conclusions drawn.

6. PLOS authors have the option to publish the peer review history of their article (what does this mean?). If published, this will include your full peer review and any attached files.

Reviewer #1: No

Reviewer #2: No

---

## [Author Response · Author response to Decision Letter 0]

15 Apr 2020

Response to Reviewers

Reviewer #1 comments: 

1. The question of how activity level influences symptom severity is important to the management of IBS symptoms.

Answer to comment 1:

We appreciate the reviewer’s helpful comments to improve our manuscript. This point is exactly our research question. Since this is a very important comment, we have added this sentence to the Introduction section. Please see page 4, lines 57-58 in the revised manuscript.

“The research question of how activity level influences symptom severity is important to the management of IBS symptoms.”

2. There were 1240 surveys distributed and 663 (53%) returned usable data. 103 of these had an IBS diagnosis and were included in the study. It is not clear whether the participation rate was influenced by the subject's knowledge of the study aims.

Answer to comment 2:

We value the reviewer for this important comment, it is better to calculate and compare the study participation rate of those who had IBS diagnosis before this survey and those who did not, but there was no data on this in this study. The results of this study may have been influenced by the information available to the participants about the relationship between gastrointestinal symptoms and physical activity. This is one of the limitation of this study. Therefore, we have added the information provided during the informed consent process to the participants in the Methods section. In addition, it was described as sixth limitation in the Discussion section.

Methods section, Study Design and Ethical Considerations subsection: P4, lines 67-71: “Participants were informed that the purpose of this study was to investigate the relationship between GI symptoms and physical activity among participants with IBS during the informed consent procedure. Furthermore, they were instructed to measure their physical activity for 1 week using a Pedometer (LifeCorder GS, Suzuken, Tokyo), and to wear the pedometer for 1 week (except when taking a bath), and also to perform their daily life activities.”

Discussion section: P12, lines 225-231: “The purpose of the study was explained to the participants during the informed consent process. Thus, participants may have increased their physical activity during the study period since they were informed that IBS symptoms are affected by the amount of physical activity. It is unclear whether the information bias was influenced by the participants’ knowledge of the study aims. Therefore, the effect of the bias should be verified by a study examining the effects of information on the relationship between GI symptoms and physical activity among people with IBS.”

3. Sex was predominantly female, with 78 females to 23 males. Sex was not a significant predictor. Should it be included in the analysis?

Answer to comment 3:

We value the reviewer for this comment. The prevalence of IBS is more common in females than in males, and being female is considered a risk factor in IBS [1, 2]．On the other hand, Bjorkman et al. reported that there was no gender difference with depressive symptom, pain, defecation frequency, effects on daily living, and dissatisfaction with bowel habits in IBS [3]. Therefore, we analyzed the relationship between gastrointestinal symptoms and locomotor activity by gender including all participants (including males). As a result, no significant fit was found in the logistic model in male (please see Results section, page 9, lines 161-165). These results may be related to gender differences in IBS pathology and differences in age and physical activity. Of course, the limitation of the analysis in this study was the small number of males. Therefore, we added this issue to the Discussion section as follows:

“There was no significant association between GI symptoms and locomotor activity in males with IBS in this study. There were gender differences in the symptoms of IBS characterized by constipation and diarrhea. The prevalence of IBS subtypes were similar to those in a previous study; no IBS subtypes were included in our analysis. In addition, symptoms vary with age; therefore, future studies with larger cohorts should be stratified by age and IBS subtype to further investigate the relationship between physical activity and digestive symptoms.” (page 12, lines 219-225:)

4. No instructions to subjects are included in the manuscript. These may influence the outcome of the data.

Answer to comment 4:

We agree with the reviewer about this comment. The results of this study may have been influenced by the knowledge of the participants about the relationship between digestive symptoms and physical activities. This is a limitation of the research. Therefore, we have added the information provided during the informed consent process to the participants in the Methods section. In addition, it was described as sixth limitation in the Discussion section.

Methods section, Study Design and Ethical Considerations subsection: P4, lines 67-71: “Participants were informed that the purpose of this study was to investigate the relationship between GI symptoms and physical activity among participants with IBS during the informed consent procedure. Furthermore, they were instructed to measure their physical activity for 1 week using a Pedometer (LifeCorder GS, Suzuken, Tokyo), and to wear the pedometer for 1 week (except when taking a bath), and also to perform their daily life activities.”

Discussion section: P12, lines 225-231: “The purpose of the study was explained to the participants during the informed consent process. Thus, participants may have increased their physical activity during the study period since they were informed that IBS symptoms are affected by the amount of physical activity. It is unclear whether the information bias was influenced by the participants’ knowledge of the study aims. Therefore, the effect of the bias should be verified by a study examining the effects of information on the relationship between GI symptoms and physical activity among people with IBS.”

5. Among the predictor variables (age, sex, symptom severity, and activity), only activity level was significantly related to symptom severity. Age was restricted because the population were students. Although none of the predictor variables was related to symptom severity, an "ordinal logistic regression" method was used to control for overlapping classification of the predictors. The logic is not clear to me and should be explained. The supporting tables suggest that the data were analyzed by changes in activity level needed to move a subject from one severity classification to another on a 5-point scale. If this method appropriately controls for the classification overlap, the analysis shows a strong effect of age and sex on the severity of IBS symptoms.

Answer to comment 5:

We appreciate the reviewer for these comments. We performed a stratified analysis of gender difference. Age was not adjusted for because all participants’ age categorized them in the younger age range, instead we described this as limitations of the study. In the title, we added "among younger people: ". In addition, a characteristic of the ordinal logistic regression model is that a linear function is applied to the logistic regression function. Equation 2 explains the f (x) part of Equation 1. Thus, we have added texts about the statistical analysis as follows:

“The relationship between GSRS and pedometer counts (prediction probability g (x)) was estimated using ordinal logistic regression modeling (Equation 1) with the dependent variable as GSRS (f (x), continuous variates 1 to 7) and the independent variable as pedometer counts for x (Equation 2). The principle of ordinal logistic regression modeling is to fit the probability (P) of multiple dichotomous responses (Equation 1):” (Methods section, Statistical Analysis subsection, page 6, lines 101-105)

In this study, the probability of the cut-off point for each level of severity of GI symptoms based on the GSRS score (1|2, 2|3, 3|4, 4|5) was evaluated in association with the pedometer count.” (Methods section, Statistical Analysis subsection, page 6, lines 109-111)

6. There are some limitations to the study which should be explained: Age was limited by studying students. Sex is predominantly female; should the males be included in the analysis? Students receiving medications for IBS were excluded; does this bias the outcome?

Answer to comment 6:

We appreciate the reviewer for these helpful comments. In the results of the ordinal logistic regression analysis performed in this study, only the factor ‘male’ was not a significant fit (z=-1.81, p=0.07, Figures), only being female was significant (z=-2.44, p=0.01), and no gender case was significant (z=-3.05, p=0.002). Please see the figure (not shown in this text) and the Results section on males. To clarify whether the gender difference was a confounding factor, male data were included in the analysis of this study. Our response to this comment of the reviewer is related to comment 4 of the reviewer. Please see our response to comment 4. In addition, since the participants in this study were young, we have changed the title to reflect the age.

7. Explain how this study extends previous studies of the effects of physical activity on IBS symptom severity. How should clinicians use the findings to manage patients?

Answer to comment 7:

We thank the reviewer for the pertinent question and agree with its relevance. Accordingly, we have added the following texts in the Discussion section:

Previous studies have shown that exercise improves IBS symptoms. Current data has suggested the "degree" (amount) of physical activity required to attenuate IBS symptoms. The results of this study can provide the clinicians with information on how many steps to add to the current physical activity level among IBS patients that can reduce GSRS by 1 point.” (Discussion section, page 13, lines 238-242)

8. Please explain the tables and figures. Are changes in activity level related to the baseline level of activity? Are the recommended changes achievable?

Answer to comment 8:

We appreciate the reviewer for these questions. This study was an observational study; gastrointestinal symptoms and pedometer counts were both obtained as baseline data. The number of steps is the data for one week, and the GSRS is the data after measuring the amount of walking. Please see revised manuscript in page 5, lines 93-96 and footnote in Table 1. In a further intervention study, it will be necessary to determine whether the gastrointestinal symptoms would improve as expected if the patient achieves the level of physical activity indicated in this study. The daily walk recommended by Health Japan 21 is 8,500 steps for a female and 9,000 steps for a male [4]. If the locomotor activity of IBS patient is low or does not reach the recommended level, according to Health Japan 21, the patient can be instructed to exercise to that extent. We added these texts in the Discussion section as follows:

“Previous studies have shown that exercise improves IBS symptoms. Current data has suggested the "degree" (amount) of physical activity required to attenuate IBS symptoms. The results of this study can provide the clinicians with information on how many steps to add to the current physical activity level among IBS patients that can reduce GSRS by 1 point.” (Discussion section, page 13, lines 238-242) 

Reviewer #2 comment:

The aspect that the study addresses is important. Although the effect exercise on IBS is well documented, this study provides a quantitative measure of effect of walking on IBS symptoms. It will be helpful for management of this disease. The manuscript is well written. I have one main question.

9. What was the reason the authors used Rome III criteria for confirming IBS instead of Rome IV criteria? Please see the following three references given as example how using criteria Rome IV versus Rome III impacts studies.

(i) Am J Gastroenterol (2018) 113:1017–1025. https://doi.org/10.1038/s41395-018-0074-z

(ii) Lin and Chang; Clinical Gastroenterology and Hepatology Vol. 18, No. 2

(iii) https://doi.org/10.1111/nmo.13189

The authors need to address this in the manuscript as it may impact the conclusions drawn.

Answer to comment 9:

We appreciate Reviewer #2 for the useful comments provided to improve our manuscript. We have added these texts with recommended references in the revised manuscript as follows: 

“This study used Rome Ⅲ criteria, when the study was planned in 2015. The IBS diagnostic criteria were updated to ROME Ⅳ, and the Japanese version of the questionnaire has not yet been published. In Rome IV, the evaluation of the symptoms has changed from that of Rome Ⅲ, including the emphasis on the subjective experience of abdominal pain in IBS patients, and the demarcation between the constipation subtype of IBS and functional constipation. In subsequent studies, IBS diagnostic criteria should be based on ROME Ⅳ.” (Discussion section, pages 12-13, lines 231-236)

References

1. Choghakhori R, Abbasnezhad A, Amani R, Alipour M. Sex-Related Differences in Clinical Symptoms, Quality of Life, and Biochemical Factors in Irritable Bowel Syndrome. Dig Dis Sci. 2017;62(6):1550-60. Epub 2017/04/05. doi: 10.1007/s10620-017-4554-6. PubMed PMID: 28374085.

2. Bijkerk CJ, Muris JW, Knottnerus JA, Hoes AW, de Wit NJ. Randomized patients in IBS research had different disease characteristics compared to eligible and recruited patients. J Clin Epidemiol. 2008;61(11):1176-81. Epub 2008/07/16. doi: 10.1016/j.jclinepi.2008.02.001. PubMed PMID: 18619799.

3. Bjorkman I, Jakobsson Ung E, Ringstrom G, Tornblom H, Simren M. More similarities than differences between men and women with irritable bowel syndrome. Neurogastroenterol Motil. 2015;27(6):796-804. Epub 2015/03/31. doi: 10.1111/nmo.12551. PubMed PMID: 25817301.

4. Nishi N, Okuda N. National Health and Nutrition Survey in target setting of Health Japan 21 (2nd edition). Journal of the Natlonal Institute of Public Health. 2012;6(5):399-408.

---

## [Decision Letter · Decision Letter 1]

1 May 2020

PONE-D-19-33098R1

The effects of locomotor activity on gastrointestinal symptoms of irritable bowel syndrome among younger people: an observational study

PLOS ONE

Dear Prof. Hamaguchi,

Thank you for submitting your manuscript to PLOS ONE. After careful consideration, we feel that it has merit but does not fully meet PLOS ONE’s publication criteria as it currently stands. Therefore, we invite you to submit a revised version of the manuscript that addresses the points raised during the review process.

We would appreciate receiving your revised manuscript by Jun 15 2020 11:59PM. To enhance the reproducibility of your results, we recommend that if applicable you deposit your laboratory protocols in protocols.io, where a protocol can be assigned its own identifier (DOI) such that it can be cited independently in the future. For instructions see: http://journals.plos.org/plosone/s/submission-guidelines#loc-laboratory-protocols

We look forward to receiving your revised manuscript.

Kind regards,

Subas Neupane

Academic Editor

PLOS ONE

Additional Editor Comments (if provided):

Few things in the methods should be clarified further.

Present briefly the Rome III criteria to diagnose IBS symptoms. Also, describe briefly the Gastrointestinal Symptoms Rating Scale (GSRS) and how it was used in the analysis.

Please describe briefly the process to obtain data from pedometer on physical activity, and how it was used in the analysis for e.g. average of all days of data or every day’s data, etc.

Reviewers' comments:

Reviewer's Responses to Questions

**Comments to the Author**

1. If the authors have adequately addressed your comments raised in a previous round of review and you feel that this manuscript is now acceptable for publication, you may indicate that here to bypass the “Comments to the Author” section, enter your conflict of interest statement in the “Confidential to Editor” section, and submit your "Accept" recommendation.

Reviewer #1: All comments have been addressed

Reviewer #2: All comments have been addressed

2. Is the manuscript technically sound, and do the data support the conclusions?

Reviewer #1: Yes

Reviewer #2: Yes

3. Has the statistical analysis been performed appropriately and rigorously? 

Reviewer #1: Yes

Reviewer #2: Yes

4. Have the authors made all data underlying the findings in their manuscript fully available?

Reviewer #1: Yes

Reviewer #2: Yes

5. Is the manuscript presented in an intelligible fashion and written in standard English?

Reviewer #1: No

Reviewer #2: Yes

6. Review Comments to the Author

Reviewer #1: 1. The authors have addressed the comments in my previous review, and the manuscript appears ready to be accepted for publication. However, there are a few minor points that should be addressed as listed below:

2. Figure 3 is identical to Figure 2, and the image described as Figure 3 in the manuscript does not exist. This oversight is easy to remedy.

3. On line 87, change the word "momentum" to "activity".

4. In lines 141 and 178, delete the word "fashion"; it is adequate without this word.

Reviewer #2: The authors addressed my question about using the Rome III criteria and revised the text and references accordingly.

7. PLOS authors have the option to publish the peer review history of their article (what does this mean?). If published, this will include your full peer review and any attached files.

Reviewer #1: No

Reviewer #2: No

---

## [Author Response · Author response to Decision Letter 1]

8 May 2020

Response to Academic Editor comments

Academic Editor comments:

Few things in the methods should be clarified further.

1. Present briefly the Rome III criteria to diagnose IBS symptoms. 

Answer to comment 1:

We appreciate the academic editor for the helpful comments to improve our manuscript. We added the text to explanation the Rome Ⅲ criteria in our revised manuscript as follows: 

“The Rome III criteria are used to diagnose IBS symptoms, which include recurrent abdominal pain or discomfort, 3 days per month in the last 3 months (12 weeks), and are associated with two or more of the following three criteria: 1) improvement with defecation, 2) the onset is associated with a change in stool frequency, and 3) the onset is associated with a change in the stool form (appearance). To fulfil the criteria, symptom onset should occur 6 months prior to the diagnosis.” (Methods section, pages 5-6, lines 92-96)

2. Also, describe briefly the Gastrointestinal Symptoms Rating Scale (GSRS) and how it was used in the analysis.

Answer to comment 2:

We agree with you regarding this comment. We have described about the GSRS and how it was used in the analysis as follows: 

“The GSRS is a disease-specific instrument of 15 items combined into 5 symptom clusters depicting reflux, abdominal pain, indigestion, diarrhea, and constipation. The GSRS has a seven-point graded Likert-type scale where “1” represents the absence of troublesome symptoms and “7” represents very troublesome symptoms.” (Methods section, page 6, lines 99-102)

“The GSRS scores were derived from the total score and divided by 15 (i.e. the 15 item subscales). The average pedometer counts (steps/day) were calculated using all days of data collection. The association between the GSRS score and pedometer counts was determined by the ordinal logistic modeling analysis [21]. The relationship between GSRS score and pedometer counts (prediction probability g (x)) was estimated using ordinal logistic regression modeling (Equation 1) with the dependent variable as GSRS score (f (x), continuous variates 1 to 7) and the independent variable as pedometer counts for x (Equation 2). (Methods section, page 7, lines 110-116)

3. Please describe briefly the process to obtain data from pedometer on physical activity, and how it was used in the analysis for e.g. average of all days of data or every day’s data, etc.

Answer to comment 3:

We agree with you on this comment. We have described briefly the process of how to obtain data from the pedometer on the physical activity, and how it was used in the analysis as follows:

“Walking activity data that were recorded in LifeCorder GS were uploaded into a personal computer using an application Lifelyzer05 (Kenz, Tokyo).” (Methods section, page 6, lines 105-106)

“The average pedometer counts (steps/day) were calculated using all days of data collection.” (Methods section, page 7, lines 111-112) 

Response to Reviewers

Reviewer #1 comments: 

The authors have addressed the comments in my previous review, and the manuscript appears ready to be accepted for publication. However, there are a few minor points that should be addressed as listed below:

1. Figure 3 is identical to Figure 2, and the image described as Figure 3 in the manuscript does not exist. This oversight is easy to remedy.

Answer to comment 1:

We appreciate the reviewer’s helpful comments to improve our manuscript. We carefully checked and replaced Figures 2 and 3. The figures are very similar and at first glance they look the same, but they are different. For example, the number of scatter plots is different, and the logistic curves are slightly different. Please see the revised Figures 2 and 3.

2. On line 87, change the word "momentum" to "activity".

Answer to comment 2:

We agree with the reviewer about this comment and have changed the word "momentum" to "physical activity" for consistency with all other uses in the revised manuscript (Methods section, page 5, line 87).

3. In lines 141 and 178, delete the word "fashion"; it is adequate without this word.

Answer to comment 3:

We agree with the reviewer on this comment and have deleted the word "fashion" in the revised manuscript (Pages 9-11, lines 151-190).

Reviewer #2: The authors addressed my question about using the Rome III criteria and revised the text and references accordingly.

Answer to comment:

We would like to thank Reviewer #2 for carefully reviewing our manuscript and for the positive support of our research.

---

## [Editor Report · Decision Letter 2]

19 May 2020

The effects of locomotor activity on gastrointestinal symptoms of irritable bowel syndrome among younger people: an observational study

PONE-D-19-33098R2

Dear Dr. Hamaguchi,

We are pleased to inform you that your manuscript has been judged scientifically suitable for publication and will be formally accepted for publication once it complies with all outstanding technical requirements.

With kind regards,

Subas Neupane

Guest Editor

PLOS ONE
---

## [Editor Report · Acceptance letter]

21 May 2020

PONE-D-19-33098R2 

The effects of locomotor activity on gastrointestinal symptoms of irritable bowel syndrome among younger people: an observational study 

Dear Dr. Hamaguchi:

I am pleased to inform you that your manuscript has been deemed suitable for publication in PLOS ONE. Congratulations! Your manuscript is now with our production department. 

With kind regards,

on behalf of

Dr. Subas Neupane 

Guest Editor

PLOS ONE